# The Case of the Kumbhalgarh Wildlife Sanctuary and Camel Pastoralism in Rajasthan (India)

**Ilse Köhler-Rollefson [1,*] and Hanwant Singh Rathore [2]**

1   League for Pastoral Peoples and Endogenous Livestock Development, Pragelatostr. 20, 64372 Ober-Ramstadt, Germany

2   Lokhit Pashu-Palak Sansthan, Sadri 306702, India; lpps.sadri1996@gmail.com

*   Correspondence: ilse@pastoralpeoples.org

**Abstract:** The Indian forest management system introduced during colonial times has led to the progressive loss of the grazing rights of the country's pastoralists, culminating in the abolishment of grazing fees and replacement with grazing fines in 2004. This scenario has had a negative knock-on effect on the conservation of many of the livestock breeds that pastoralists have developed in adaptation to local environments and that are the basis of the country's food security. This paper illustrates the dilemma with the example of the Kumbhalgarh Wildlife Sanctuary (KWS) in Rajasthan that represents the traditional monsoon grazing area for local camel, sheep and goat pastoralists. Raika herders have engaged in a long-standing but losing legal battle with the state for their continued seasonal access to this area. This situation contributes to the rapid decline of the camel which is an iconic part of Rajasthan's desert identity, a major attraction for tourists and was declared state animal in 2014. The aims of the forest department to conserve wild animals and those of pastoralists and camel conservationists could easily be integrated into a more equitable governance system as is endorsed by Aichi Target 11 of the CBD Strategic Plan 2011–2020. However, deeply engrained concepts about nature being separate from (agri-)culture, as well as unequal power structures, stand in the way.

**Keywords:** Kumbhalgarh Wildlife Sanctuary; agrobiodiversity; Raika; pastoralism; camels; fortress approach

## 1. Introduction

India prides itself on being a mega-biodiversity country and has established 104 National Parks and 566 Wildlife Sanctuaries, which together cover about 5% of its geographical area [1]. India is also equipped with more than a proportionate share of livestock breeds, amounting to approximately 6% of global domestic animal biodiversity [2].

In recent decades, conservation concepts have evolved from an earlier fortress approach to the recognition that successful governance of protected areas requires consultation and active involvement of all stakeholders, including the communities that depend on the conserved areas for their livelihoods. Equity in conservation respecting the rights of actors was endorsed by the CBD COP14 in 2018, and during the COP 15, the U.N. special rapporteur on human rights and the environment stated that indigenous-led, rights-based conservation is the only way forward [3].

One of the main stakeholder groups in protected areas in India are its diverse groups of pastoralists who live and work throughout the country [4]. They have traditionally depended on forest grazing for at least part of the year, especially during the monsoon season. Herding groups that include the Gaddi and the Van Gujjars in the Himalayas, the Banni buffalo breeders of Kutch in Gujarat, the Toda and various cattle breeding groups in Tamil Nadu, and many others have been affected by the establishment of protected areas and sought to draw attention to the situation over the last decades, including at the COP 11 of the CBD held in Hyderabad in 2015 [5,6]., This communication focuses on the

governance of the Kumbhalgarh Wildlife Sanctuary (KWS) and the role and position of one of its main stakeholder groups, the Raika pastoralists (Figure 1).

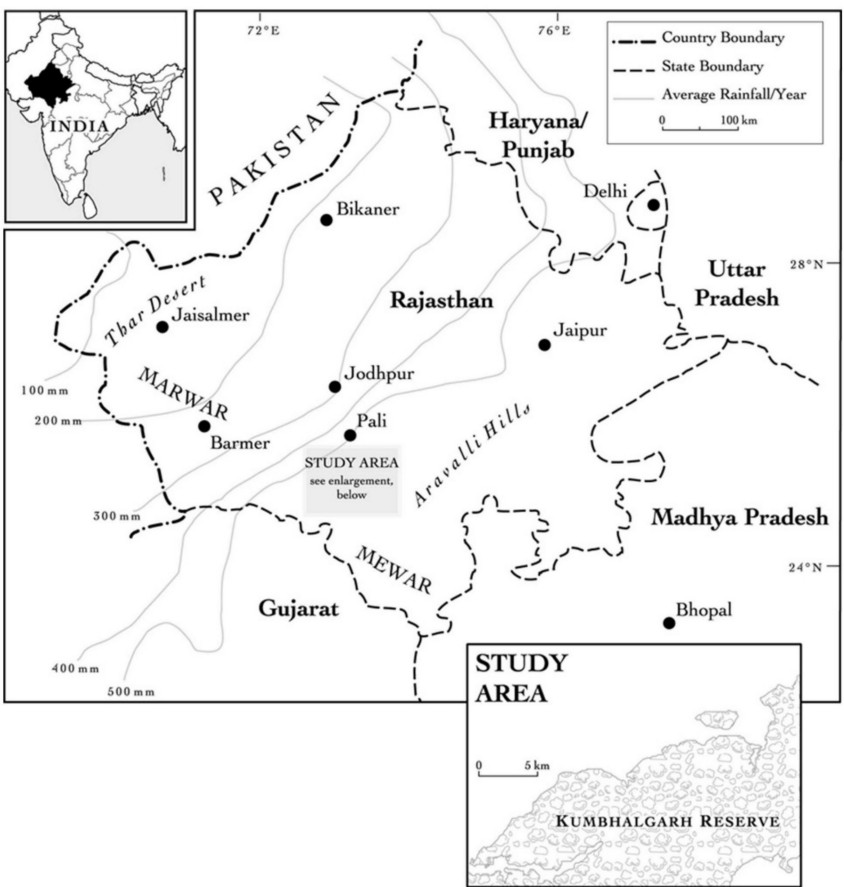

**Figure 1.** Location of the Kumbhalgahr Protected Area in South-Central Rajasthan (India. Credit: Paul Robbins).

The Kumbhalgarh Wildlife Sanctuary located in south-central Rajasthan is one of 25 such protected areas in Rajasthan and was notified in 1971. It covers 610.53 km square km along 85 km of the western face of the Aravalli Hills and extends over parts of three districts (Pali, Udaipur, Rajsamand) in south-central Rajasthan. The sanctuary represents an ecotone between the Thar Desert and the Aravalli Hills and is home to the Indian wolf (*Canis lupus*), Indian leopard (*Panthera pardus fusca*), sloth bear (*Melursus ursinus*), striped hyena (*Hyaena hyaena*), Golden jackal (*Canis aureus*), jungle cat (*Felis chaus*), sambhar (*Rusa unicolor*), nilgai (*Boselaphus tragocamelus*), chausingha (*Tetracerus quadricornis*), chinkara (*Gazella bennettii*) and Indian hare (*Lepus nigricollis*).

The Raika, also known as Rebari, are the largest pastoralist group of Western India. They herd sheep, goats, and cattle, and have an intimate relationship with camels. They are Hindu caste and believe that their primeval forefather was created by God Shiva for taking care of camels. Due to this divine connection, they felt responsible early on for the welfare of the camel and have observed several taboos in their interaction with it, such as not selling female camels to anyone outside the community, never slaughtering camels or eating their meat, and never selling their milk. The only 'product' they traditionally could monetize was male camels at the annual livestock fairs, of which Pushkar is the most famous [7].

The traditional management system in this semi-arid to sub-humid part of Rajasthan consists of village-based or nomadic herding, with camels feeding on a mosaic of fallow fields, village grazing grounds and the forest. They are well integrated with crop cultivation, and there are long-standing relationships between Raika and farmers to manure their fields.

Farmers traditionally compensated herders in kind or with cash for staying on their fields overnight and depositing dung [8].

*Raika Tradition of Stewardship for Biodiversity*

The Raika are well known as stewards of agricultural biodiversity for their role as creators of a number of livestock breeds including Marwari and Boti sheep, Sirohi goat, Nari cattle and as guardians of the camel and holders of extensive indigenous knowledge, as they have documented in their Biocultural Community Protocol [9]. They are known for not cutting trees, except lopping them during certain parts of the year, and being extremely equanimous about losing livestock to predators, such as leopards, to the extent of not claiming compensation for such losses. In fact, small livestock, such as sheep and goats represents a major part of the diets of leopards, as is known from scat analysis. The Raika also put out forest fires and repair waterpoints that are used by wildlife. The community has documented their role in biodiversity conservation and aspects of their traditional knowledge in the Raika Biocultural Community Protocol (Figure 2). They have contributed to the establishment of the Community Protocol of the Camel Breeders of Rajasthan [10]. Development of Community Protocols are a pre-scribed procedure under the Nagoya Protocol on Access to Genetic Resources and the Fair and Equitable Sharing of Benefits Arising from their Utilization, under the Convention on Biological Diversity (CBD). Raika representatives have participated in a number of UN level meetings, such as the Conferences of the Parties (COPs) to the Convention on Biological Diversity (CBD).

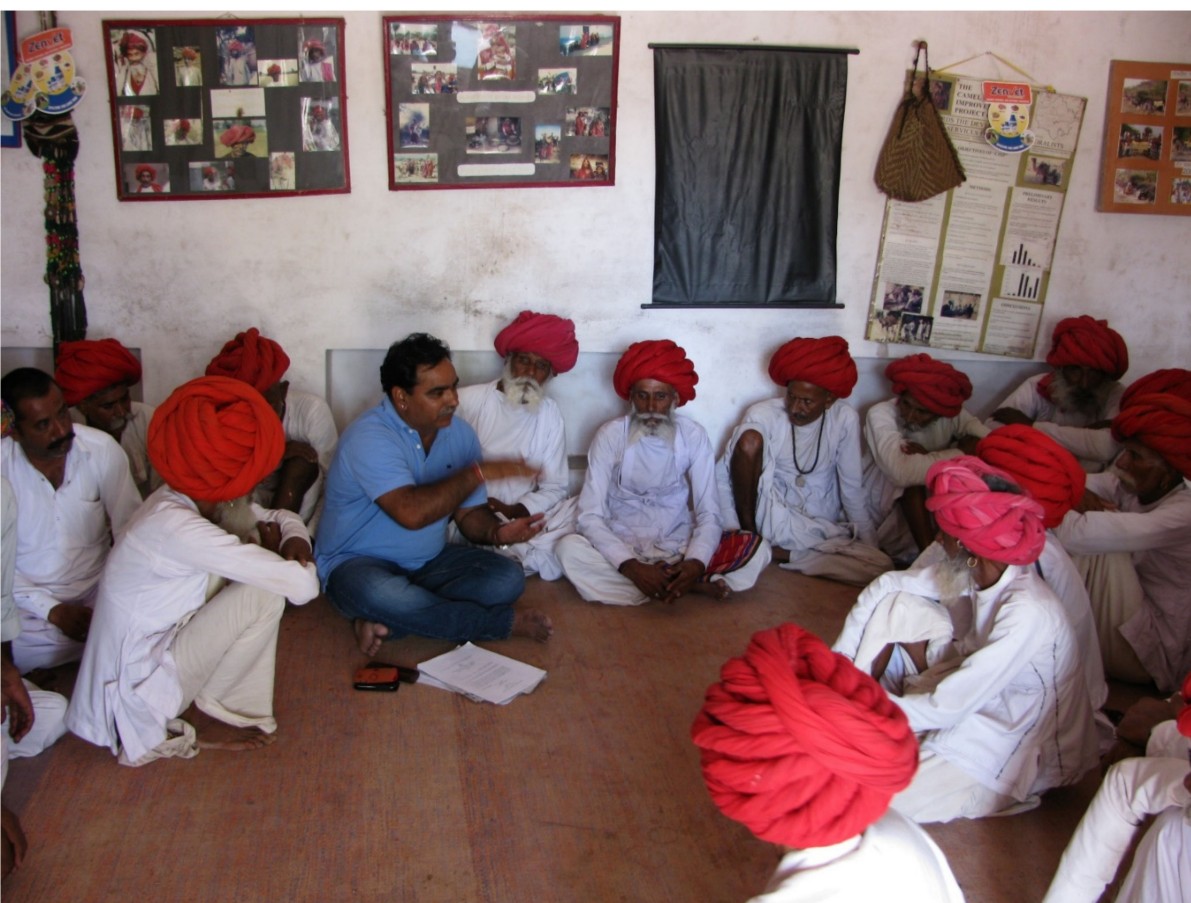

**Figure 2.** Raika elders in 2009 discussing the concept of a Community Protocol under the Nagoya Protocol on Access to Genetic Resources and the Fair and Equitable Sharing of Benefits Arising from their Utilization in which they assert their role as guardians of the camel and creators of agrobiodiversity in the form of various local livestock breeds. The engagement of pastoralists and other stakeholder communities represents an important step towards equity oriented conservation and a required element of good governance.

India's livestock breeds have been developed by communities based on indigenous knowledge and breeding practices [11]. They are adapted to specific environments and production systems. Ruminant livestock plays an important part in maintaining soil fertility and indigenous livestock breeds raised in agro-pastoral systems provide around 50% of the nation's milk and more than 70% of its meat [8]. These herding systems make use of a mosaic of common property resources (CPR), including fallow land, village grazing grounds, 'revenue land' and also forests [12]. In fact, forests represent a major source of livestock nourishment, especially during the monsoon seasons when fields are cultivated.

Eviction of livestock from protected areas and forests represents one of the major reasons for the decline of local livestock breeds, together with increasingly restricted access to other types of CPRs [12]. One of India's most iconic livestock types is the one-humped camel, which was designated as the state animal of Rajasthan in 2014, a move which has resulted in a further decline of the species, with numbers declining by 34.69% between 2012 and 2019, due to the implementation of a law that interfered with making a livelihood from camel breeding, as outlined below.

## 2. Materials and Methods

The socio-economic and legal developments around the Kumbhalgarh Wildlife Sanctuary were observed over the 30-year period from 1991 to the present, first in the context of a research project on camel management practices by one of the authors (IKR), later through a series of support and advocacy projects for Raika pastoralists who depend on access to the forest during the monsoon season when agricultural fields are cultivated. These projects were conducted under the auspices of the NGO Lokhit Pashu-Palak Sansthan (LPPS) led by the second author (HSR) and set up in response to appeals for veterinary help by the Raika during initial fieldwork The size of the local camel population using the KWS for monsoon grazing was monitored by means of household surveys in irregular intervals. Observations in the field were complemented with a review and analysis of legal documents pertaining to court cases for grazing rights that had been initiated with the help of the NGO.

## 3. Results

### 3.1. The Demise of the Camel and Its Status as State Animal of Rajasthan

The one-humped camel (Camelus dromedarius) is a symbol of Rajasthan that stands for its desert ecology and exerts a big draw for both domestic and international tourists. Historically, it was used for transportation through the Thar desert, initially as a pack animal, but as a cart animal since the 1960s. In the 1980s, India had a camel population of around 1.5 million; currently it is in the range of 200,000 camels. While the global camel population has doubled in the last half century, in India it has experienced a steep decline, due to several reasons. According to the camel herders, disappearance of grazing resources due to exclosure from forests, fencing, and irrigated agriculture are the prime causes for the decline of camel herding, followed by lack of veterinary care. Herders see a connection between camels being undernourished and prevalence of diseases. However, another major cause is the replacement of camels with tractors and trucks, leading to a lack of demand for them at livestock fairs. Thus, the number of camels on the Pali side of KWS reduced by 60–70% between 1995 and 2013, and in 2013, the number of camels depending on forest grazing was estimated at between 300–550 [8]. Currently it is estimated at around 150, based on records kept by the Kumbhalgarh camel dairy.

When India's 19th Livestock Census revealed in October 2012 that the Indian camel population had declined by 22.48 per cent over a five-year period and was down to around 300,000 in Rajasthan, the government became alarmed and, hoping to remedy the situation, declared it as 'state animal' on 30 June 2014. This move perplexed the camel herders and the general public, as it was not clear what this implied. In order to implement this status and because of a strong push by animal welfare groups, the Rajasthan Camel (Prohibition of Slaughter and Regulation of Temporary Migration or Export) Bill was passed by the

Legislative Assembly on 27 March 2015, although the Raika had cautioned against this move in several letters and representations. This law prohibited the export of camels from Rajasthan across state borders as well as the use of camels for meat. It did not address the factors that had been identified by the community as causing the decline; the disappearance of camel grazing areas was not addressed. The 20th Livestock Census conducted in 2019 revealed a further decline by 34% since 2012 [13].

### 3.2. State Attitude towards Pastoralism

In pre-colonial India, peasants and pastoralists co-existed in the landscape, playing their different ecological roles in a mutually supportive way, with pastoralists providing draught animals and organic fertilizer, which they exchanged with farmers for grain. This was fundamentally changed during the colonial era, when pastoralists came to be regarded as lazy, unproductive and unlawful, even notified as criminal tribes [14]. According to European notions at the time, land that was not cultivated was considered wasteland and it was deemed their duty to bring it under control and to make it fertile. Initially, forests were included in the wasteland category, but when colonizers realized their commercial potential, they separated them into various categories according to their perceived economic value. Pastoralists were seen as destroying forest productivity (as were other traditional users) and undermining their commercial value. The India Forest Act of 1865 turned forests into state property and the rights of the people who had used them traditionally for food, fuel and forage were nullified [15]. Even today, the dependence of India's enormously productive livestock sector on access to forests and other CPRs in what essentially represent agro-silvo-pastoral systems is not recognized [4].

### 3.3. History of the Kunbhalgarh Wildlife Sanctuary and Community Legal Challenges

The changing rules that govern the use of the area that composes the Kumbhalgarh Wildlife Sanctuary illustrate how herders have gradually been deprived of their customary rights and how there has been little if any change from the colonial mindset (Figure 3).

Until the early 20th century, the area below the Kumbhalgarh Fort was a favourite hunting area of the Maharajah of Jodhpur. Local pastoralists had grazing privileges on land known as '*gudara*'. In 1884, the Assistant Conservator of Forest Ajmer-Marwar examined forests for their commercial value and in 1887, the proprietorship of the forest was transferred to the state. Being aware of the economic importance of pastoralism, the Maharajah made sure the Raika retained grazing privileges. In 1963, the grazing fee per camel was officially set at Rs 5/head.

In 1971, the Kumbhalgarh forest was declared as protected area under Rajasthan Wild Animals and Birds Protection Act 1951 to conserve the habitat of sloth bear, leopard, wolf, hyena, sambhar and grey jungle fowl.

In 1990, the Chief Conservator of Forest and Wildlife upheld grazing rights, and, in 1998, the Collector of Pali district ascertained that rights given at the time of the constitution of forest blocks would continue. But in 1999, the Forest Protection Committee of Rajasthan banned grazing in the Kumbhalgarh Wildlife Sanctuary. In 2002, a writ petition was filed in the Rajasthan High Court on behalf of the Raika requesting to not be stopped from grazing their animals in KWS, and in 2003, the Rajasthan High Court permitted Raika from adjoining villages to graze.

In 2004, the Central Empowered Committee (CEC, a committee of experts constituted by the Supreme Court in 2002) requested 'strict compliance of the Hon. Supreme Court's order to not allow removal of dead, dying, diseased trees and grass from any national park and sanctuary'. In response, the Rajasthan Forest department stopped issuing grazing permits, contrary to the High Court order of 2003 that had upheld Raika grazing rights.

Following this, the Raika Sangarsh Samiti (Raika Struggle Society) requested clarification from the CEC. As no response was received, it filed a writ petition in Rajasthan High Court requesting the State Government to issue grazing permits. This was disposed of,

with reference to the Central Empowered Committee's letter of 2004, but gave liberty to seek appropriate clarification in Supreme Court.

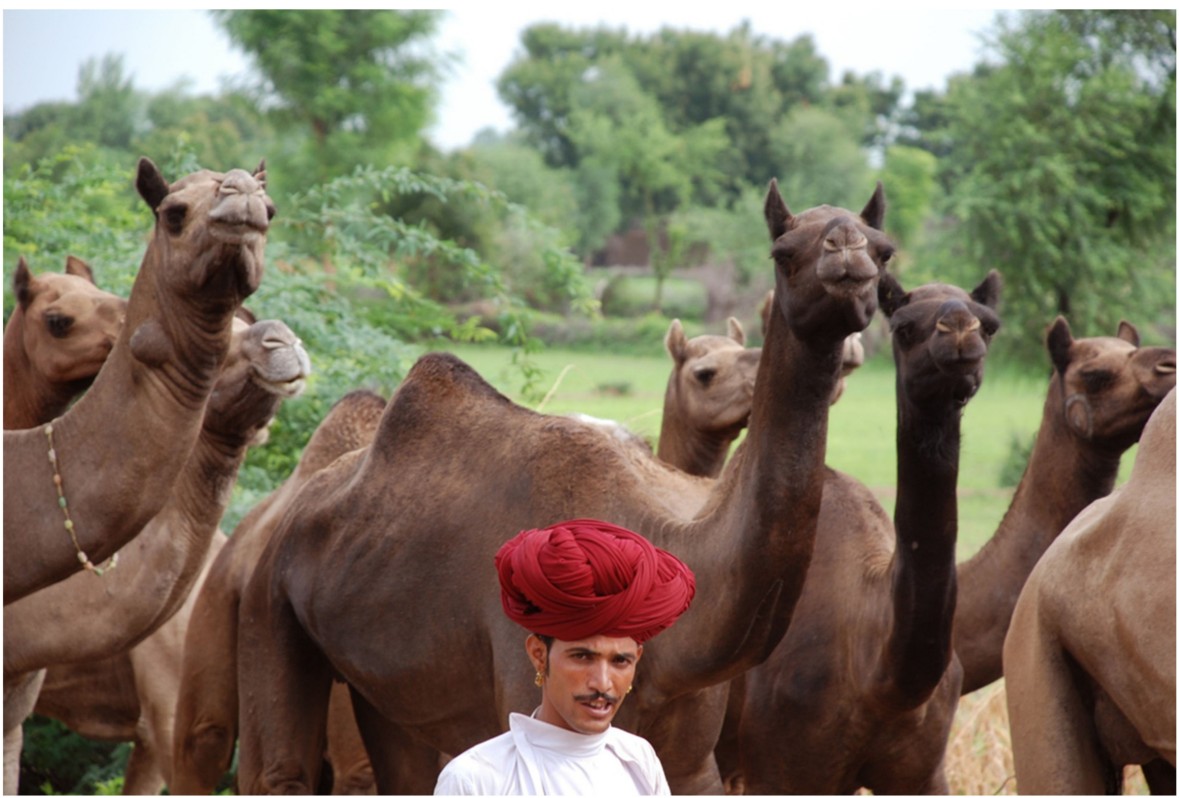

**Figure 3.** Bhanwarlal Raika is one of the very few young Raika maintaining an interest in camel breeding. Most others have been discouraged by the loss of customary grazing rights in the Kumbhalgarh Wildlife Sanctuary, among other factors. Without secure grazing rights, the 'integrated social-ecological system' of the Raika and the associated extensive knowledge system about holistic conservation is unravelling with repercussions for the conservation of agrobiodiversity and hence long-term food security.

Thus, on 22 February 2006, Raika application in Supreme Court asks for a clarification about ban on traditional grazing rights of Raika in KWS as held in order of 26.3. 2003, stating that unless grazing rights were given, '*the Raikas as a cultural and ethnic group and the camels they breed will cease to exist in near future.*'

In response, the Chief Wildlife Warden filed a report on behalf of the State of Rajasthan stating ' . . . *in order to protect one of the last remains of Aravalli biodiversity, it is recommended that grazing should not be permitted in the KWS area.*'

Due to a lack of resources, the Raika withdrew their case at the advice of their legal counsel.

Meanwhile, also in 2006, the Indian Parliament had enacted the Scheduled Tribes and Other Traditional Forest Dwellers (Recognition of Forest Rights) Act, 2006 (FRA) that had the purpose of correcting some of the historic injustice to forest dwelling communities and to provide use and access rights to people who have traditionally used forests and included a specific reference to pastoralists who use the forests on a seasonal basis, declaring them eligible for receiving such rights. This development again raised hopes among the Raika, and as members of village committees, they submitted a number of claims for Community Forest Rights. Unfortunately, although these claims were filed, they were never processed by the authorities.

In 2012, it was announced that KWS would be turned into a National Park, which led to massive protests by the peripheral villages located in Pali district. The issue was shelved with a change in state government, but after five years, in 2019, when the previous

government returned, KWS was again notified as National Park, with 41 villages in Pali district formally raising objections to the District Collector. Once again, the issue became dormant, but in 2020, it became known that the area around KWS was to be gazetted as Eco-Sensitive Zone. Currently, the National Tiger Conservation Authority explores the viability of KWS as tiger reserve, although tiger experts regard the area as not suitable for this species [16]. At the same time, there is a renewed push for implementation of the Forest Rights Act.

## 4. Discussion

In the last couple of decades, the importance of governance and the need to address the social impacts of conservation have been widely acknowledged. They were first highlighted at the IUCN World Parks Congress in 2003 and elaborated on in the Convention of Biological Diversity (CBD) Programme of Work on Protected Areas in 2004 [4]. More recently, Aichi Target 11 of the CBD Strategic Plan 2011–2020 stated that protected areas should be equitably managed. During the recent CBD COP15, there were strong calls for rights-based approaches (RBA) in conservation [4,17].

However, the legal tussle around grazing rights in the KWS illustrates that attitudes towards local communities have not substantially changed since colonial times. According to the CBD's paragraph 8j, each contracting party shall, *as far as possible and as appropriate and subject to national legislation, respect, preserve and maintain knowledge, innovations and practices of indigenous and local communities embodying traditional lifestyles relevant for the conservation and sustainable use of biological diversity and promote their wider application with the approval and involvement of the holders of such knowledge, innovations and practices and encourage the equitable sharing of the benefits arising from the utilization of such knowledge innovations and practices*. In their community protocols, the Raika have amply put on record their role and credentials as holders of traditional knowledge that is relevant for the conservation and sustainable use of biological diversity, but this has not helped them in being considered as partners in conservation. Although formally Joint Forest Management is practiced in Rajasthan, this has not led to more collaborative or community-based approaches to forest management that have gained currency among conservationists. Nor are there any efforts for scientific research to observe and monitor the impact of livestock grazing on wild biodiversity.

The fact that monsoon grazing in the KWS is now illegal has discouraged the continuation of herding activities by the communities that traditionally depended on it. This development pertains not only to camel pastoralism but also to keepers of sheep and goats, and it is equally impacting farmers who have aspirations for organic farming, as manure is becoming a scarce resource. Eventually the agro-ecological integrity of the area surrounding the KWS may unravel.

Although the alienation of customary grazing areas is not the sole driver of the decline of the camel in Rajasthan, it is a crucial factor, and the one experienced most immediately by Raika camel herders, many of whom now have set their hopes on an emerging market for camel milk as an opportunity for saving their ancestral herds. Alternative scenarios can easily be imagined, but for these to be envisioned and eventually implemented, the antagonistic relationship between herders and government authorities would need to be overcome and trust be established.

## 5. Conclusions

Tracing the history of the area that is currently known as the Kumbhalgarh Wildlife Sanctuary illustrates the gradual loss of rights of a community that had developed knowledge and practices for combining food production with biodiversity conservation over many generations, skills that are very much in demand in an era known as Anthropocene where agriculture writ large is the biggest driver of biodiversity loss. The seeds for the current fortress approach to wildlife conservation were laid in colonial times and unfortunately, there has been no effort to decolonize the state institutions that are tasked with protecting

the environment. There is an urgent need now for adopting more people-centered concepts of conservation building on still existing ethics of wildlife protection among the people living around KWS, especially its pastoralists.

The motivations behind the most recent twist in the saga of the KWS to re-introduce tigers are not clear, except that it is a global flagship species that ensures conservation funds and income from tourists. An alternative approach would be to examine the possibility of focusing efforts on the conservation of the Indian wolf, a species of which there are smaller numbers than of tigers in India and that may be more threatened [18]. Since the occurrence of the wolf is closely linked to the presence of sheep and domestic livestock, this could lead to an innovative community-based conservation model in which the local pastoralists and agro-pastoralists are involved and that could prove to be a trail-blazing model in India for people-centered conservation. In such a way, trade-offs between the conservation of wildlife and of domestic animal diversity could be minimized, as well as negative repercussions on both rural livelihoods and national food security.

It is urgent that the new conservation paradigms calling for more equity, and as elaborated in the Aichi targets and called for during COPs 14 and 15, are integrated into the governance structures of protected and conserved areas in India. This would benefit both conservation and livelihood goals while also contributing to the future food security of the country.

**Author Contributions:** Writing—review and editing, I.K.-R.; project administration, H.S.R. All authors have read and agreed to the published version of the manuscript.

**Funding:** Research was funded through a series of fellowships to IKR from the American Institute of Indian Studies in 1990/91 as well as the German Research Council and the Alexander von Humboldt Foundation in 1996 and 1996. An on-going series of projects to support Raika pastoralists and implemented by the NGO Lokhit Pashu-Palak Sansthan (LPPS) under the directorship of HSR provided the opportunity for longitudinal data collection and was/is supported by Misereor since 1996.

**Institutional Review Board Statement:** Not applicable.

**Informed Consent Statement:** Informed Consent was obtained from all subjects involved in the study.

**Data Availability Statement:** The legal documents that are referred to are available and accessible in the archives of Lokhit Pashu-Palak Sansthan.

**Acknowledgments:** We are grateful to the various funding organizations that have made this work possible and appreciative of the countless Raika pastoralists that have engaged in the struggle.

**Conflicts of Interest:** The authors are co-founders of the Kumbhalgarh Camel Dairy, a social enterprise which seeks to conserve the camel by providing income for Raika herders. The funders of the projects on which this research is based had no role in the design of the study; in the collection, analyses, or interpretation of data; in the writing of the manuscript, or in the decision to publish the results.

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
