# Peer review of "The Case of the Kumbhalgarh Wildlife Sanctuary and Camel Pastoralism in Rajasthan (India)"

_sustainability, doi:10.3390/su132413914_

Round 1

Reviewer 1 Report

This is an interesting commentary on the history of the tussle between state conservation authorities and traditional livestock breeders and herders. My suggestion to the authors is to better reference their work in the rich literature on state/pastoralist conflict, especially with reference to protected areas, and access to grazing rights. This will then allow them to situate their narrative in a broader context, and identify instances where similar problems were faced by other groups, and the possible solutions that arose from them, or the end result of such conflict.

This will also allow a more structured format to emerge. The way the manuscript is currently structured does not really fit in to the formal methods results discussion template. It should really be a more flowing social science narrative, without these artificial sections.

the authors should also consider adding more statistics on camel production, as well as references/data on the flora and fauna of this region, and if any deleterious impacts occurred due to the stoppage of grazing rights from the early 2000s onwards.

  Some possible refs (not an exhaustive list) Saberwal, Vasant K. "Pastoral politics: Gaddi grazing, degradation, and biodiversity conservation in Himachal Pradesh, India." Conservation Biology 10, no. 3 (1996): 741-749. Saberwal, Vasant K. "Conservation as politics: wildlife conservation and resource management in India." (2000): 166-173. Kothari, Ashish, Saloni Suri, and Neena Singh. "People and protected areas; rethinking conservation in India." The ecologist 25, no. 5 (1995): 188-195. Agrawal, Arun, and Elinor Ostrom. "Collective action, property rights, and devolution of forest and protected area management." In Collective Action, Property Rights, and Devolution of Natural Resource Management. Exchange of Knowledge and Implications for Policy. Proceedings of the International Conference held from, pp. 21-25. 1999.

Author Response

We agree that the communication could be improved by adding references to the various publications suggested.

We also agree that the formal structure is somewhat in the way of a more free-flowing narrative, however we adhered to it because it was the one given.

References to the flora and fauna in the area can be given, as well as on camel production, however data on the impacts of the loss of grazing rights are not really available.

Reviewer 2 Report

This communication describes the case of the Raika pastoral caste of northwest India, its historical stewardship of agro-genetic diversity, and the decline of its herds owing to exclusion from key historically accessed resources owing to exclusionary conservation zeal. As such, the short piece covers a lot of critically important ground.

First, of course, it provides a clear, single-reading summary of the specific challenges facing this pastoral cate, one famous worldwide, crucial for preserving India’s dwindling camel populations, and a not -insignificant contributor to Rajasthan’s tourism sector. While documentation of the situation of the Raika already exists, this represents a clear, brief, and urgent synopsis.

But the paper does a great deal more work. Notably, it points towards the problem more generally of the gazetting of national parks without consultation or concern for key stakeholders, especially pastoralists. In India, this is all too common and represents a true challenge to equitable sustainability. I would urge the authors, having made the point that this case is of many, to insert a footnote that might point readers to similar problems faced by pastoral communities elsewhere in India (and there are many). Such a reference might better reinforce that the case does not stand alone but represents a class of problems with which the Indian state needs to wrestle more realistically.

Beyond this, the paper actually points to conservation alternatives that might be more sustainable, specifically including the use of the Kumbhalgarh reserve as a conservation site for wolves, a species in desperate need of conservation attention and one more fully-aligned with livestock production (ironically). Here a well, a reference or two reinforcing these points might bolster the case.

One question I do have, which probably isn’t worth major revision, is that the abstract points to the case showing the myriad ways that “deeply engrained concepts about nature being separate from (agri-)culture, as well as unequal power structures” stand in the way of reform. That point isn’t explicitly reiterated in the text, so far as I can tell. It could be.

In sum, the issue of pastoralism and conservation in India (and indeed South Asia more generally) is one of paramount socioecological significance.

This is an urgent communication and appropriate for Sustainability.

Author Response

Thank you for the supportive comments.

We agree, it is important to add a foot note or reference similar situations experienced by pastoralists in other places in India - which is all over the country.

Thanks for pointing out the discrepancy between the abstract and the text. We will try to resolve that.

This manuscript is a resubmission of an earlier submission. The following is a list of the peer review reports and author responses from that submission.